# Application of Artificial Neural Networks in Construction Management: A Scientometric Review

**Hongyu Xu** [1]**, Ruidong Chang** [2]**, Min Pan** [3]**, Huan Li** [1]**, Shicheng Liu** [1]**, Ronald J. Webber** [4]**, Jian Zuo** [2] **and Na Dong** [1,*]

1   College of Architecture and Environment, Sichuan University, Chengdu 610065, China; xuhongyu@stu.scu.edu.cn (H.X.); lihuan@stu.edu.cn (H.L.); liushicheng@stu.edu.cn (S.L.)
2   School of Architecture and Built Environment, The University of Adelaide, Adelaide 5005, Australia; ruidong.chang@adelaide.edu.au (R.C.); jian.zuo@adelaide.edu.au (J.Z.)
3   Sichuan Kaiyuan Engineering Project Management Consulting Co., Ltd., Chengdu 610041, China; sckyzx@sckyzx.com
4   Department of Mining-Built Environment, Central Queensland University, Rockhampton 4701, Australia; r.webber@cqu.edu.au
*   Correspondence: dongna@scu.edu.cn; Tel.: +86-85407049

**Abstract:** As a powerful artificial intelligence tool, the Artificial Neural Network (ANN) has been increasingly applied in the field of construction management (CM) during the last few decades. However, few papers have attempted to draw up a systematic commentary to appraise the state-of-the-art research on ANNs in CM except the one published in 2000. In the present study, a scientometric analysis was conducted to comprehensively analyze 112 related articles retrieved from seven selected authoritative journals published between 2000 and 2020. The analysis identified co-authorship networks, collaboration networks of countries/regions, co-occurrence networks of keywords, and timeline visualization of keywords, together with the strongest citation burst, the active research authors, countries/regions, and main research interests, as well as their evolution trends and collaborative relationships in the past 20 years. This paper finds that there is still a lack of systematic research and sufficient attention to the application of ANNs in CM. Furthermore, ANN applications still face many challenges such as data collection, cleaning and storage, the collaboration of different stakeholders, researchers and countries/regions, as well as the systematic design for the needed platforms. The findings are valuable to both the researchers and industry practitioners who are committed to ANNs in CM.

**Keywords:** artificial neural network (ANN); construction management; scientometric analysis; future trends

## 1. Introduction

The characteristics of high investment, long period and high uncertainty make construction management an indispensable part of the modern construction industry [1,2]. The urgent need of upgrading and transformation of the construction industry also drives the renewal of construction management concepts and methods [3–5]. In this data-intensive industry, data, which can significantly improve the performance of CM, is becoming the key resource [6,7]. Nevertheless, data application in CM has been considered relatively conservative [1]. It is difficult to analyze and process the large volume data of the construction industry with traditional technologies, so that a large amount of data is shelved and wasted [8]. The digitization report released by McKinsey indicated that the construction industry is one of the worst-performing digitally at the moment, which maybe is the main reason for decades of persistently low productivity in the construction industry [9,10]. Therefore, in the digitalization era, it is of great significance for the construction industry to use intelligent technology to process large volume data in CM and obtain knowledge



hidden in the data for assist decision making [11]. Furthermore, one of the most promising technologies is the artificial neural network (ANN) [12].

ANN, as a mathematical model inspired by and imitating the biological brain, can be used to extract knowledge hidden in large historical data, and to process it productively [13]. As an importance branch of artificial intelligence, largely due to its good self-learning, self-organizing function and high-speed computing ability, ANN does not need to assume the relationship between variables and performs well in dealing with complex nonlinear problems [14,15]. Even with incomplete or previously unknown data acceptable results can be obtained [16]. Accordingly, ANN is extremely suitable for dealing with practical CM problems that are difficult to solve by mathematical methods and traditional modeling [13] and has been used in the Architecture, Engineering and Construction industry (AEC) since the early 1990s [17]. Previous studies have shown that ANN can play a major role in prediction, optimization, classification, and decision-making in CM practice [18–20]. It has successfully aided in solving specific problems throughout the project's life cycle from the planning stage to the operation and maintenance stage.

Because of the considerable use of ANN in CM [12], the literature related to ANN has proliferated and several available literature reviews on ANN in AEC have been put forward. For instance, Rajesh systematically reviewed the literature related to ANN in energy analysis [20], and Sony et al. [21] reviewed the related applications of convolution neural network (CNN) from the perspective of structural state assessment. However, these two literatures on ANN had specific perspectives, which only focused on the energy or the structural state assessment. Pan et al. [22] provided a comprehensive review on AI in construction engineering and management in which ANN is only mentioned to a limited extent in some paragraphs throughout the review, rather than in terms of a detailed vision. Adeli [23] reviewed the literature on the application of ANN in CM published from 1989 to 2000, but now this work is limited by timeliness. It is not difficult to conclude that the existing literature on ANNs in CM has rarely been comprehensively and systematically reviewed in the last 20 years. If the literature review of ANN in CM is not updated, this may lead to the following problems. Firstly, due to the lack of a comprehensive review, it is difficult for the beginner to learn about the authoritative authors, outlets, publications and the active countries/regions to serve as an example. Secondly, a comprehensive application profile is needed to show current research progress including what topics have been focused on and what progress has been made in the field of ANN in CM. Finally, without a summary of research evolution and current breakthroughs and limitations, researchers interested in this field will spend more time ascertaining current research status, future trends and possible research directions.

Therefore, it is necessary to make a comprehensive and systematic review of the application of ANN in CM. This paper intends to achieve the following objectives: (1) Identify main research authors, institutions and countries/regions that are active in the field of ANN in CM over the past 20 years and their cooperation relationships; (2) Present the main research interests on ANN in CM over the past two decades; (3) Uncover the relationships among different research interests and their evolution tendency; (4) Summarize benefits and challenges of ANN in CM and propose promising research directions.

## 2. Materials and Methods

Different methods are available for reviewing literatures [24] among which the scientometric analysis is good at visualizing significant structure and trends based on author, keyword, and reference in a large body of literature data [25]. The scientometric analysis can meet all of the research objectives mentioned above. Therefore, it is adopted in this paper. A three-stage process was carried out including: data collection, scientometric analysis, and discussion and conclusion. The outline of research methodology is shown as Figure 1.

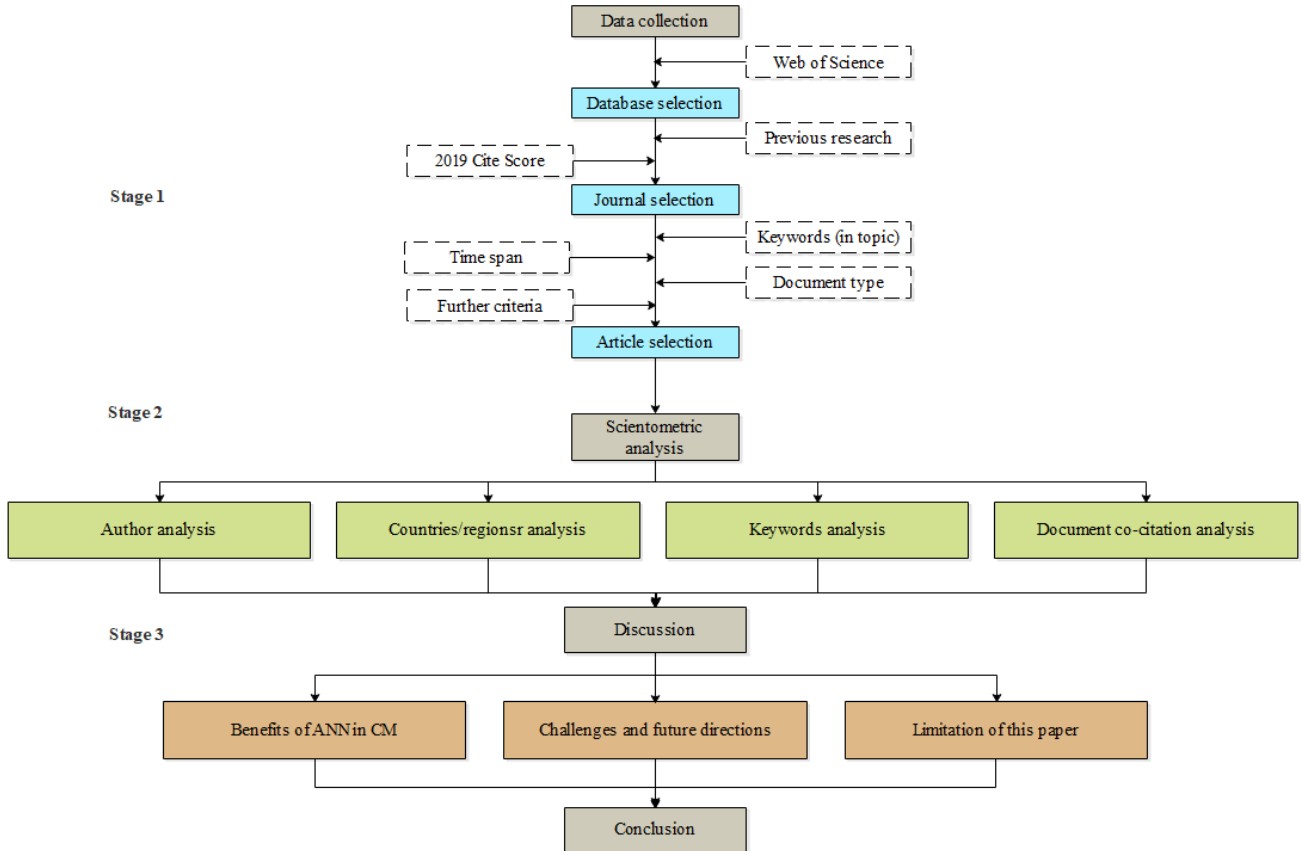

**Figure 1.** Outline of research methodology.

### 2.1. Data Collection

Due to the interdisciplinary nature of CM, a comprehensive academic database, Web of Science, known for its comprehensiveness, organizational structure, and scientific robustness, was chosen in this study [26]. It is a consensus in the industry that articles in high-ranking journals usually have high influence [25]. Considering this fact, the authors selected the journals that have an important impact and top quality in CM based on the 2019 Scopus journal metrics (CiteScore is not less than 2) and the ranking of CM journals [27]. Moreover, in view of timeliness, these journals must have published at least three papers related to ANN in CM between 2000 and 2020.

To summarize the research progress in the past 20 years, the retrieval time of the journal article is limited to 2000 and 2020. In order to ensure that all relevant papers are captured, different search keywords are applied. The following retrieval code was adopted, and the search was conducted using the 'topic' in the Web of Science.

(neural network) AND (construction management), (neural network) AND (engineering management)

Four hundred potential articles were initially identified and then filtered based on the criteria that ANN emerged as the main technology or played an important role rather than just a comparison. A two-stage selection strategy was adopted to meet the above criteria. Firstly, title, abstract, and keywords in each article are examined to exclude unrelated articles. Secondly, the entire paper content is analyzed in detail to ensure that all the selected articles are closely related to the research objectives. Finally, 112 papers were selected for later scientometric review; the different journals in which the selected papers were published can be seen in Table 1. These articles provide a representative sampling of existing studies on the ANN in CM and form the dataset utilized in the current research.

**Table 1.** Distribution of the selected papers among different journals.

| No. | Journal | Cite Score 2019 | Literature List | No. of Paper |
|---|---|---|---|---|
| 1 | Journal of Construction Engineering and Management (JCEM) | 5.8 | [15,28–64] | 38 |
| 2 | Automation in Construction (AC) | 9.5 | [65–98] | 34 |
| 3 | Journal of Civil Engineering and Management (JCiEM) | 4.7 | [99–111] | 13 |
| 4 | Engineering, Construction and Architectural Management (ECAM) | 2.5 | [112–119] | 9 |
| 5 | Journal of Management in Engineering (JME) | 6.7 | [120–127] | 8 |
| 6 | International Journal of Project Management (IJPM) | 13.0 | [128–133] | 6 |
| 7 | Journal of Computing in Civil Engineering (JCCE) | 7.6 | [134–137] | 4 |
| total | | | | 112 |

*2.2. Introduction and Process of Scientometric Analysis*

The earliest definition of scientometrics is "quantitative research on scientific development research" [138]. The purpose of scientometric analysis is to help literature reviews overcome subjective issues in content analysis [139]. A scientometric analysis consists of the text-mining and citation analysis which helps researchers find systematic literature-related findings, by finding literature information that may be ignored in manual review studies [140]. There are several available tools to realize the goal of scientometric analysis such as VOSviewer and Citespace. Citespace is an advantageous application for analyzing and visualizing networks, and is specialized in analyzing what the major research interests are and how they are evolved and linked [141]. VOSviewer offers the basic functionality required for producing, visualizing, and exploring bibliometric networks, and also has special text-mining features [142]. Each tool has its own strengths, and it is necessary to appropriately use different tools for different kinds of analysis [143]. The combination of these two tools is widely accepted for reviews, such as that of AI in the Architecture, Engineering and Construction industry [12] and computer vision applications for construction [144]. Therefore, VOSviewer and Citespace were selected for scientometric analysis in this paper. VOSviewer was used to implement keyword co-occurrence analysis and Citespace was used for other tasks.

Whether VOSviewer or Citespace, its main process can be summarized as follows: (1) importing literature data from WOS into software for visualization; (2) Figures presentation and optimization of different aspects according to software functions; (3) In-depth scientometric analysis of the adjusted figures. After the screening mentioned above, each piece of data is downloaded from WOS as a full-record text format. In total, 112 valid publications were extracted. Then, the output file format after renaming is 'download_*.txt' and is imported into VOSviewer and Citespace for format conversion. Visualizations are generated and can be converted to different views through different functions of the software. After adjusting different visualizations of input data to make them easier to read and analyze, the scientometric analysis begins, including the following four aspects considered in this paper. First, through co-author network analysis, core research groups and their cooperative relationships were identified. Second, with network analysis of participating countries/regions, the most influential countries/regions which are particularly active on ANN in CM and the collaborations among them were described. Third, taking note of the keywords, network analysis is conducted to discern the main research interests and the hot topics on ANN in CM. Fourth, with the timeline visualization and citation bursts, the keywords evolution network shows the trends and changes. Finally, network analysis of the co-citation references was carried out to mine the most representative literature in this filed.

**3. Results of Scientometric Analysis**

*3.1. Author Analysis*

Co-authorship network analysis of current research in ANN in CM can promote access to specialists and expand research productivity [25]. The core group of authors and

their cooperative relationships in this field can be determined by analyzing the structural characteristics of the corresponding authors and their cooperation networks. In this paper, the co-authorship network was generated in Citespace and the collaboration map is presented in Figure 2. The size of the node indicates the number of the articles published by the author, and the connecting line indicates the collaboration relationship among them, and the color of the line indicates the authors in the same cluster. Publication dates from past to present are shown in a transition from cool to warm color. As can be seen from Figure 2, there are 253 nodes, 287 links, and the network density is 0.0101. The typical value for network density is between 0 and 1. Especially low network density, even close to 0, indicates that the authors in the network are not closely connected [145]. The most productive author on ANN in CM was *MINYUAN CHENG* with 8 articles, followed by *HSINGCHIH TSAI* (4), and *XUEFENG ZHAO* (4). It can be noted that many authors tend to collaborate with a relatively stable group of collaborators, so there are several major groups of authors. Among them, the cluster with *MINYUAN CHENG* is the core research team. They represent the important scholars in the application of ANN in CM and can offer highly individualized scientific research information to other researchers in this field.

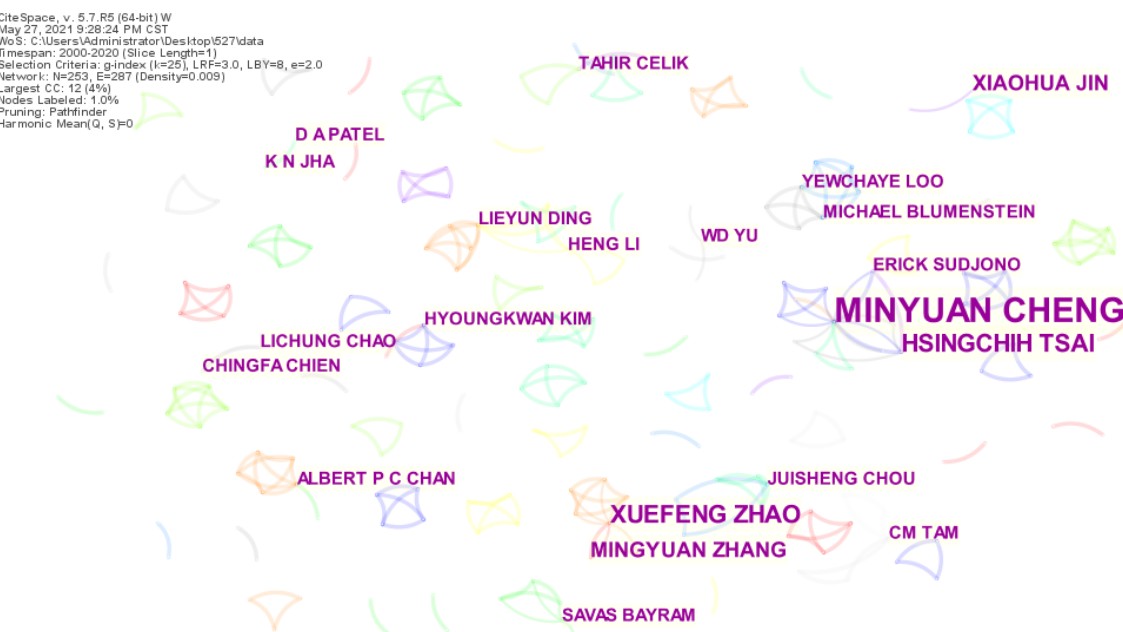

**Figure 2.** Network of co-authorship in research on ANN in CM [15,28–137].

### 3.2. Countries/Regions Analysis

The leading countries/regions in research on ANN in CM can be identified through network analysis. The results are useful for interested scholars to help them identify leading countries with high potential for cooperation. At the same time, the results can also provide top management with macro data to facilitate policy decisions on industry digitization. This section presents the countries/regions contributing to the 112 research articles extracted for the study. Figure 3 shows the network of citing countries/regions, which contains 29 nodes and 33 links. The size of a node represents the total number of published articles in the 112 articles, and the thickness of the links indicates the levels of the cooperative relationships. As a result, PEOPLES R CHINA (23 articles), TAIWAN (21 articles) and USA (17 articles) top the list, demonstrating that the considerable number of related articles in these countries/regions have made significant contributions to research in this field. However, compared with other emerging technology such as AI and BIM, as a growing new technology, ANN in CM has not yet attracted the global attention it deserves. It is believed that in the future, more and more countries/regions will pay attention to and promote research in this field. The betweenness centrality is an important index in

Citespace. Freeman [146] noted that the betweenness centrality could be calculated by the ratio of the shortest path between two nodes to the sum of all such shortest paths. The greater the betweenness centrality, the higher its importance. From the perspective of centrality, Citespace identified a collaborative pattern, and the network reveals that countries such as PEOPLES R CHINA (centrality = 0.43), USA (centrality = 0.35) and TURKEY (centrality = 0.13) were the key infrastructure nodes in the network. Researchers from these countries collaborate more actively than others. The centrality for TURKEY is only 0.13 which indicates insufficient collaboration, while Taiwan, as the second prolific region, has a centrality of 0.00. In general, these results imply that strengthening academic exchanges and contacts to expand current research productivity may be a subject worth more attention.

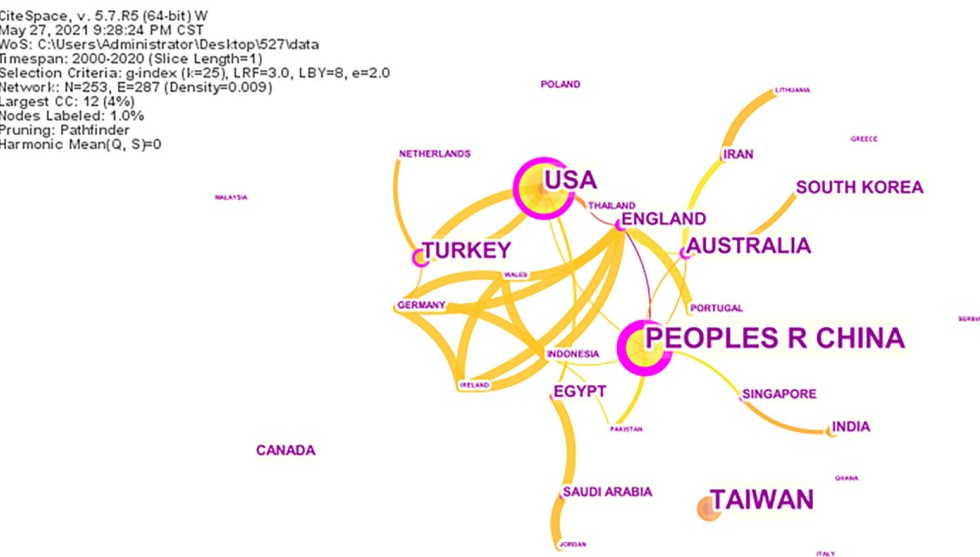

**Figure 3.** Collaboration network of countries/regions in the research on ANN in CM.

*3.3. Keywords Analysis*

3.3.1. Co-Occurrence Network of Keywords

Keywords are representative and concise descriptions of research paper content, and analyzing keywords provides an opportunity to identify major research interests in this field [147]. A network of keywords offers a good picture of a knowledge domain, which help to identify the interests over a specific timespan and provides an understanding of how they are connected and organized [138]. Identical terms (e.g., cost estimate, construction cost estimation and cost prediction; genetic algorithm and GA; regression and regression analysis) were merged (as cost estimate, genetic algorithm and regression analysis, respectively) and generic keywords related to research areas, etc. (e.g., management, model) were omitted during analysis because they do not reflect the current related research trend and have an impact on the clustering accuracy of analysis results [148].

To construct and map the knowledge domain on ANN in CM, keyword co-occurrence in the research area was obtained using VOSviewer. Main research interests on ANN in CM is shown as Figure 4. The network of co-occurring keywords has 122 nodes, 342 links, and a total link strength of 432. In this network, each node represents a different keyword, and the link between the two nodes is the co-relationship between the connected keywords, the node size is determined by the frequency of the words appeared in the 112 articles.

The frequency with which keywords are cited indicates the main research interests in the research field [149]. Table 2 summarizes the keyword occurrences and node strength of each. The links are the number of linkages between a given node and others, while the total link strength reflects the total strength linked to a specific item.

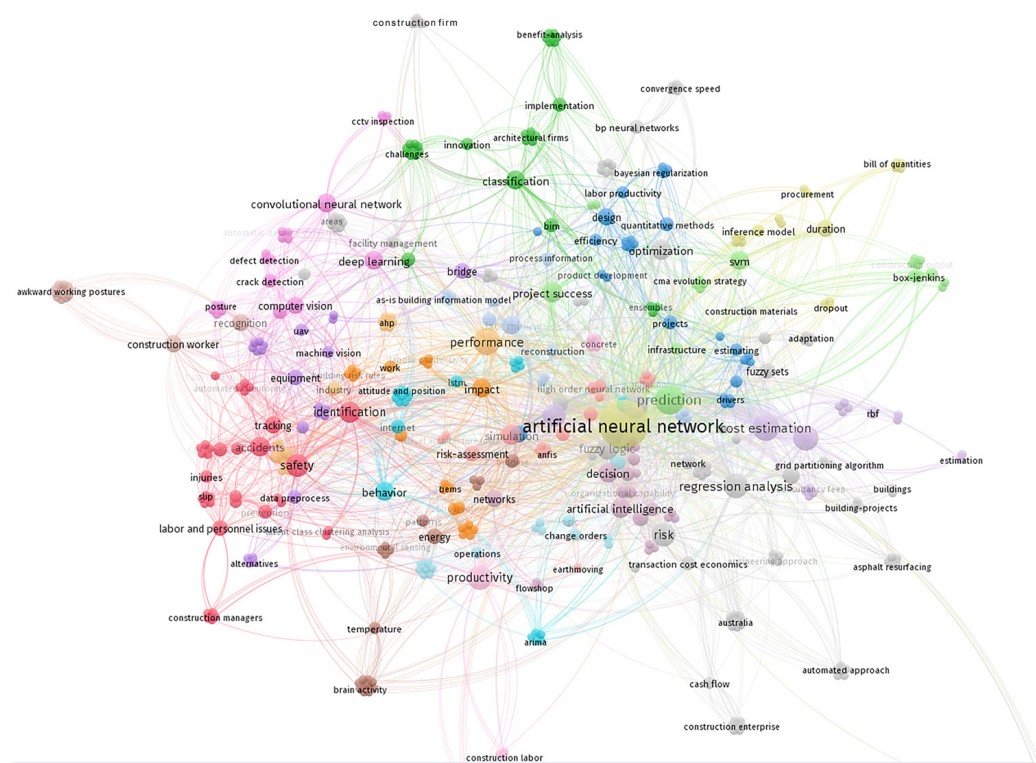

**Figure 4.** Main research interests on ANN in CM (co-occurrence network of keywords).

**Table 2.** Top keywords of existing research interests on ANN in CM.

| Keyword | Occur-rences | Links | Total Link Strength | Keyword | Occur-rences | Links | Total Link Strength |
|---|---|---|---|---|---|---|---|
| artificial neural network | 75 | 333 | 556 | data mining | 6 | 33 | 38 |
| prediction | 20 | 113 | 157 | deep learning | 6 | 59 | 66 |
| cost estimation | 16 | 79 | 127 | impact | 6 | 49 | 55 |
| performance | 16 | 102 | 138 | svm | 6 | 48 | 55 |
| construction cost | 14 | 70 | 101 | computer vision | 5 | 31 | 41 |
| genetic algorithm | 13 | 78 | 115 | construction worker | 5 | 50 | 58 |
| regression analysis | 13 | 73 | 106 | cost and schedule | 5 | 29 | 41 |
| productivity | 12 | 86 | 102 | design | 5 | 45 | 48 |
| risk | 11 | 77 | 99 | duration | 5 | 25 | 29 |
| algorithm | 10 | 80 | 96 | machine learning | 5 | 45 | 48 |
| safety | 10 | 72 | 106 | networks | 5 | 35 | 40 |
| fuzzy logic | 9 | 42 | 71 | recognition | 5 | 54 | 60 |
| identification | 9 | 63 | 88 | tracking | 5 | 53 | 61 |
| optimization | 8 | 57 | 71 | bridge | 4 | 17 | 19 |
| project success | 8 | 50 | 62 | cluster-analysis | 4 | 28 | 33 |
| simulation | 8 | 62 | 75 | contractor | 4 | 28 | 31 |
| accidents | 7 | 50 | 75 | data | 4 | 34 | 39 |
| artificial intelligence | 7 | 37 | 49 | disputes | 4 | 26 | 31 |
| decision | 7 | 54 | 70 | energy | 4 | 42 | 47 |
| behavior | 6 | 51 | 67 | equipment | 4 | 43 | 49 |
| cbr | 6 | 32 | 45 | fuzzy sets | 4 | 18 | 23 |
| classification | 6 | 42 | 45 | labor and personnel issues | 4 | 37 | 43 |
| convolutional neural network | 6 | 48 | 54 | risk-assessment | 4 | 34 | 38 |

From Table 2, it is revealed that ANN was the keyword with the highest frequency, and was used as the keyword in 75 of 112 articles, which further verifies the rationale of the literature selection. Besides ANN, prediction is the most frequent keyword and the total link strength is 113. Prediction is in the highest level of the keywords indicating the strong inter-relatedness between ANN and prediction. The analysis result that prediction has received considerable attention could be interpreted by the fact that as a powerful algorithm for AI, ANN is an effective tool for prediction [150]. ANN is typically applied in prediction models for knowledge discovery from a large quantity of information and documents which are generated in the process of construction management, and the result can provide reliable assistance for decision-making [117] and optimization [33]. In addition, ANN can also be used for recognition and classification, such as defect classification [85] and construction activity recognition [87]. Compared with prediction, research on recognition and classification attracts relatively insufficient attention and deserves further exploration in the future.

Except for the main functions of ANN in CM, the range of problems or tasks ANN has been applied to solve in CM is another important issue. It can be seen from Figure 4 and Table 2 that cost estimation, performance, productivity, risk, safety, project success and duration represent other important types of nodes in the network, which are key tasks of construction management [151]. The results indicate that ANN has gradually indeed become an effective tool for CM and is gradually replacing the traditional mainstream methods due to its advantages [79]. It is worth mentioning that behavior, construction worker, contractor, labor and personnel issues are an emerging type of research topic, and the importance of personnel management is further highlighted in project management [59]. There is, however, a conspicuous absence of interest in the topic of environment in the network, which needs further attention.

Finally, there are many keywords related to algorithms such as genetic algorithm, regression analysis, algorithms, fuzzy logic, data mining etc. It indicates that in order to better complete project management tasks, a variety of methods have become more widely applied together with ANN to improve the efficiency and precision of the model.

### 3.3.2. Timeline Visualization and Citation Bursts of Keywords

Cluster analysis is used to identify the semantic themes hidden in the textual data. Figure 5 shows a timeline visualization of cluster analysis of keywords which was created by Citespace 5.7.R1. There are three text mining algorithms that can be used to label clusters in CiteSpace. Log-Likelihood ratio (LLR) clustering technique was implemented in this study because of its good clustering results [80]. The network is divided into 9 major co-citation clusters (with cluster IDs #0, #1, etc.). CiteSpace automatically selects a label for each cluster based on titles, keywords, and abstracts of the articles in each cluster. Usually, the Modularity (Q value) and Mean Silhouette (MS) value are used to evaluate the clustering effect. Generally speaking, Q value is within the interval of [0, 1). Q > 0.3 means that the community structure is significantly divided. A cluster's silhouette value ranges from −1 to 1 and assesses the uncertainty involved in defining the cluster's nature. A value of 1 signifies that the cluster is perfectly isolated [12]. As shown in Figure 5, the network has high modularity (Q = 0.581 > 0.3), which shows that the network is divided into clusters with dense links amid nodes within clusters. The MS value is 0.363, indicating that the homogeneity of the clusters is not high. This MS value shows that although the studies in the network in each cluster may be consistent in exploring somewhat similar issues, they address different issues in fact [12].

The largest cluster (#0) has 25 members and is labeled as 'machine learning' by LLR, which includes deep learning, modeling, LSTM, machine learning, framework, hybrid intelligence, back-propagation neural network, prediction, etc. This result means that the enhancement and optimization of ANN is one of the most popular research interests in recent years and is in line with earlier observations; development of ANN-based deep learning (DL) model is a representative direction. DL affords a machine learning technique

in which computers are taught to perform what comes naturally to humans by training [150]. Zhou, Xu, Ding, Wei and Zhou [98] combined a wavelet transform noise filter, CNN, and long short-term memory predictor to propose a DL method. The DL method proposed by Rafiei and Adeli [56], including unsupervised deep Boltzmann machine learning and BPNN. The actual data is used to verify the proposed DL algorithm, and the results show that the effectiveness and accuracy of a single ANN are improved.

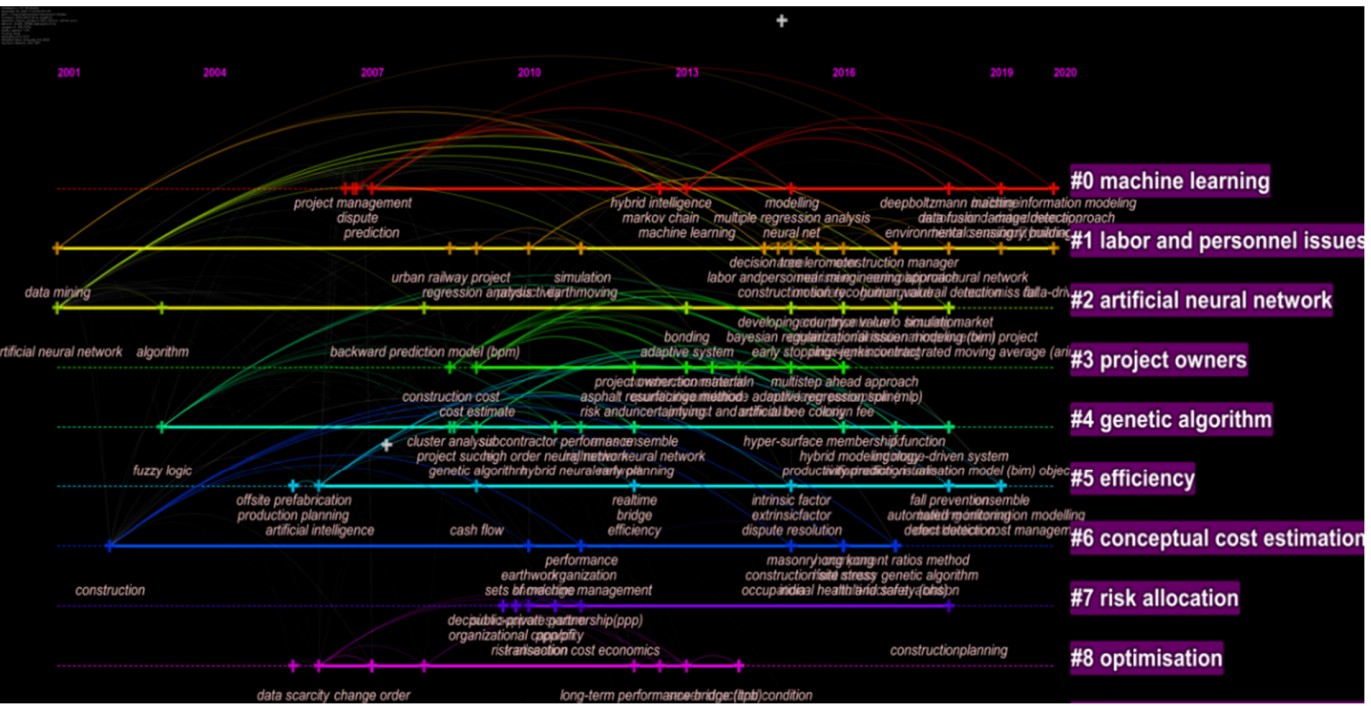

**Figure 5.** Timeline visualization of keywords (clustering structure).

The second largest cluster (#1), 'labor and personnel issues', mainly includes research on management, behavior, safety climate, worker, health, etc. The clustering results show that worker safety is another important topic. Especially since 2015, there has been a node explosion in Figure 5 (#1). This result shows that CM has paid great attention to safety and ANN has been widely used in the safety field in recent years. Emerging technologies such as laser scanning and smart sensors have made massive data acquisition a reality, which has provided tremendous support for this development [65]. Yi, Chan, Wang and Wang [94] proposed a system which could be automated by integrating smart sensor technology, location tracking technology and ANN to protect the wellbeing of those who have to work in hot and humid conditions.

In addition, cost estimation (#6) and risk allocation (#7), as the main tasks of traditional building management, were important topics before 2017, but the attention has been gradually weakened in recent years. Optimization (#8) is labeled as the smallest cluster indicating that the research on ANN applications in this area needs to be strengthened, as shown in Figure 5.

Citation bursts reflect the dynamics and evolution of the field by citing articles with a sharp increase in citations [152]. The higher the suddenness of a keyword, the more attention is paid to it in the time interval considered, and to some extent, it represents the research frontier and hotspot in the subject area [153]. Figure 6 shows the top 25 keywords with strongest citations bursts from 2000 to 2020. The light green line indicates the range of literature years reviewed, while the orange line indicates the duration of a citation burst event.

As shown in Figure 6, keywords with citation bursts can be divided into two categories including methods and issues. As to the methods appearing in the last 20 years, data mining

first burst between 2001 and 2006, followed by fuzzy logic, case-based reasoning, genetic algorithm, and regression analysis. Meanwhile, hybrid neural network, machine learning and artificial intelligence began to appear in the last 10 years indicating that as an important representative of machine learning and artificial intelligence, ANN is attracting more and more attention. ML had the strongest of the citation bursts (3.77) and the bursts began from 2018 up until 2020 implying that ML based on ANN represent emerging themes in research on ANN in CM.

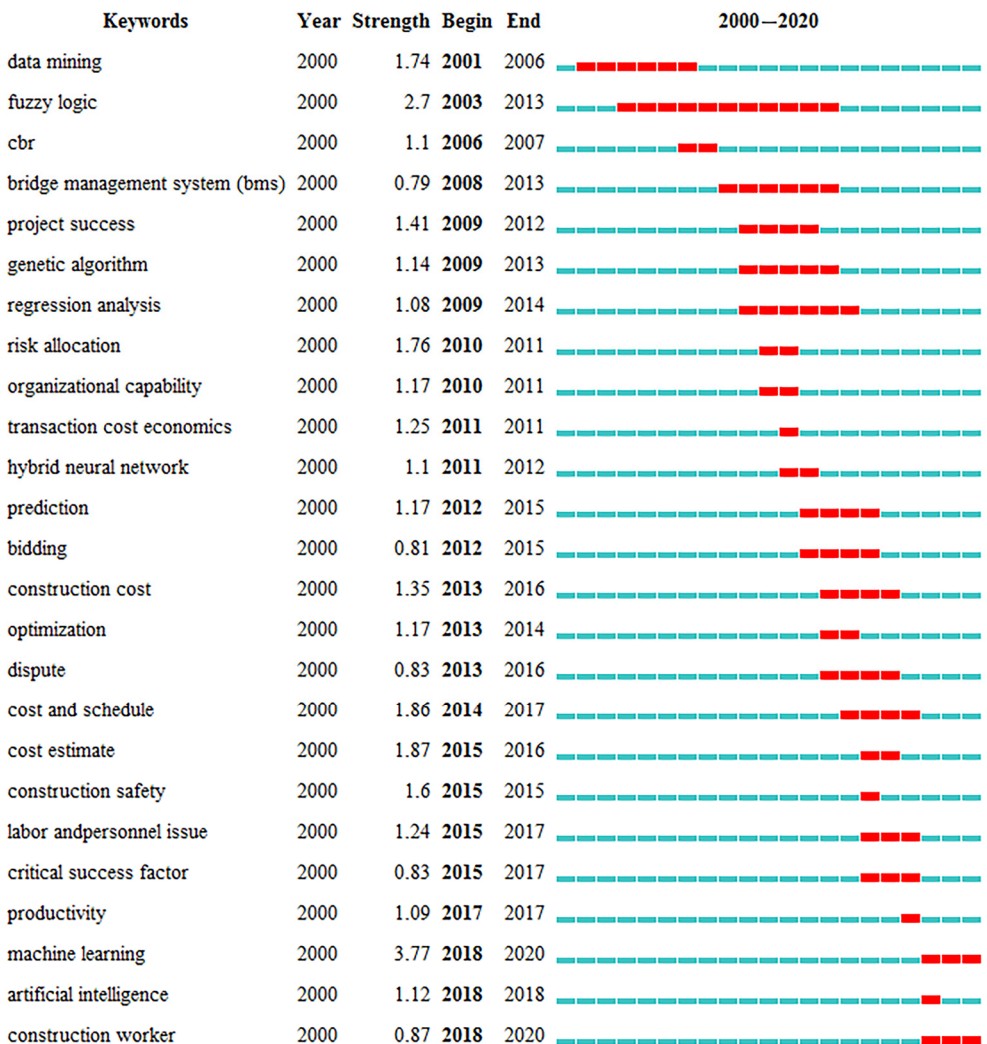

**Figure 6.** Top 25 keywords with the strongest citation bursts in the selected literature.

As to the issues, ANN has been applied to bridge management, project success, risk allocation, organizational capability, transaction and construction cost, bidding, dispute, safety, workers and productivity, etc., which have been hot research topics in the past 20 years. Before 2010, risk allocation had strong citation bursts due to the continuous development of the PPP model [134]. After 2010, Figure 6 shows that cost estimate (burst strength, 1.87), cost and schedule (1.86), had strong citation bursts in the literature. This implies that these were hot topics in the respective years.

### 3.4. Document Co-Citation Analysis

The references of frontier manuscripts can represent the knowledge base in a field [154]. Document co-citation analysis (DCA) studies a network of co-cited references. Thus, through analyzing the co-cited references, DCA objectively explored the underlying knowledge base of the ANN research in CM. CiteSpace was used to analyze the documents

cited in 112 records. Figure 7 shows the detailed outcome of the document co-citation analysis, i.e., a co-citation network including 497 nodes and 1485 links. Each link represents the co-citation relationship between the two corresponding articles while the font size represents the co-citation frequency of these documents. The node documents were among cited documents and were not necessarily included in the 112 retrieved articles.

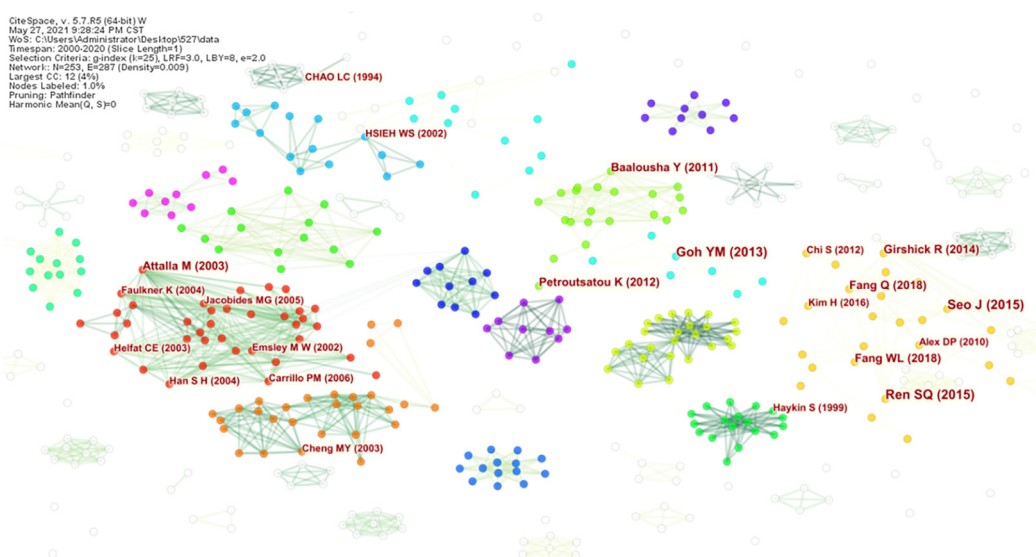

**Figure 7.** Document co-citation network of ANN in CM [15,28–137].

The top 10 cited documents are listed in Table 3. These articles were widely recognized by peers and had high value for research on ANN in CM. A systematic review of these 10 high-quality articles reveals the following findings: (1) Except for two studies on the improvement of CNN methods, the remaining research topics are divided equally into cost and safety for CM. (2) It can be seen from the publication dates of top cited documents that cost and safety have been the main research interests of the last two decades. Cost boomed in the last decade, but now the focus has turned to safety.

Four important cost documents cover almost all the research directions of ANN. First, the research on ANN: for example, articles compare traditional prediction methods (regression analysis, etc.) and ANN [155], develop different types of ANN algorithms (MLFN and GRNN) [55], other algorithms (FL) improve ANN [35], and establish a database for ANN [99]. Second, research on different topics: for example, articles include the cost for different types of construction projects, such as road tunnel construction cost [55], and the cost of reconstruction projects [155]. In addition, there is discussion of price, total cost, maintenance cost and other cost predictions from different perspectives.

For safety, the top cited literature shows ANN is mainly used for object safety detection and accident analysis. Specific techniques such as object detection, tracking and action recognition can be used to effectively identify unsafe acts and conditions. A large number of related researches in computer vision technology provide conditions for CNN to further realize accurate object detection. Fang Q developed a CNN model for automatic non-hardhat-use detection technology [156]. Fang WL proposed an improved and faster approach with CNN features to detect the presence of workers and equipment in real-time [157].

**Table 3.** Top 10 highly co-cited papers.

| No. | Author | Article | Topic | Year | Total Citations | Source |
|---|---|---|---|---|---|---|
| 1 | Seo, et al. [158] | Computer vision techniques for construction safety and health monitoring | safety | 2015 | 5 | Advanced Engineering Informatics |
| 2 | Ren, et al. [159] | Faster R-CNN: towards real-time object detection with region proposal networks | CNN | 2017 | 5 | IEEE Transactions on Pattern Analysis and Machine Intelligence |
| 3 | Goh, et al. [160] | Neural network analysis of construction safety management systems: a case study in Singapore | safety | 2013 | 5 | Construction Management and Economics |
| 4 | Petroutsatou, Georgopoulos, Lambropoulos and Pantouvakis [55] | Early cost estimating of road tunnel construction using neural networks | cost | 2012 | 4 | Journal of Construction Engineering and Management |
| 5 | Girshick, et al. [161] | Rich feature hierarchies for accurate object detection and semantic segmentation | CNN | 2014 | 4 | Conference on Computer Vision and Pattern Recognition |
| 6 | Fang, Ding, Zhong, Love and Luo [157] | Automated detection of workers and heavy equipment on construction sites: A convolutional neural network approach | safety | 2018 | 4 | Advanced Engineering Informatics |
| 7 | Fang, Li, Luo, Ding, Luo, Rose and An [156] | Detecting non-hardhat-use by a deep learning method from far-field surveillance videos | safety | 2018 | 4 | Automation in Construction |
| 8 | Baalousha and Celik [99] | An integrated web-based data warehouse and artificial neural networks system for unit price analysis with inflation adjustment | cost | 2011 | 4 | Journal of Civil Engineering and Management |
| 9 | Attalla and Hegazy [155] | Predicting cost deviation in reconstruction projects: artificial neural networks versus regression | cost | 2003 | 4 | Journal of Construction Engineering and Management |
| 10 | Cheng and Ko [35] | Object-oriented evolutionary fuzzy neural inference system for construction management | cost | 2003 | 3 | Journal of Construction Engineering and Management |

## 4. Discussion

### 4.1. Benefits of ANN in CM

Specific applications of ANN in dealing with CM problems were represented by the above literature analysis. The specific benefits of ANN in CM should be summarized so as to have a clear understanding of what value has been achieved.

Firstly, compared with traditional CM, data-driven CM based on ANN makes the construction and management activity more intelligent. As a powerful instrument to intelligently discover hidden knowledge from the mass of accumulated data in completed construction projects, ANN visualizes the tacit knowledge in the project experience and provides reliable advice for automated analysis and decision-making [162]. Table 2 proves

that ANN has been successfully applied to the intelligent solution of problems such as prediction, success, cost, performance, productivity, risk and safety throughout the whole life cycle of construction projects without too much manual intervention. For instance, ANN has realized intelligent decision-making by quickly predicting the key indicators of project success in the project planning stage [75]. In the construction phase, current ANN models can provide real-time performance evaluation [49] and dynamic monitoring of on-site operations [15], so as to provide early warning. In order to ensure that the projects can be accomplished successfully, such an intelligent monitoring system is undoubtedly a great boon for on-site project managers. The intelligent maintenance system based on ANN has realized the automatic and remote assessment of structural condition [89]. In general, the intelligent function with ANN in CM is gradually replacing the traditional manual-led pattern, which tends to be time-consuming, subject to personal judgment and experience, and prone to error.

Secondly, improving the efficiency of CM is another prominent benefit of ANN being applied. Accurate estimation and reliable optimization are provided by the ANN model to make the project more effective and smoother. Experience has proved that the predictive performance of ANN is indeed better than traditional methods such as multiple regression analysis [120], and the recently optimized ANN model has made great progress in prediction performance [97]. Reliable results can avoid potential errors and reduce unnecessary waste, which is essential to the success of the construction project. The optimization effect is reflected in generating valuable suggestions for improvement in complex CM tasks under conflicting requirements and limitations, such as cash flow control [74], capital allocation plan [101], optimization control of schedule plan [69], time allocation of material processing [67], construction quality inspection [136], and adjustment of construction machinery posture and position [98]. Such findings can guide the optimization of the construction execution process, enabling timely adjustments at an early stage. Therefore, unnecessary steps, re-working, conflicts and potential delays can be effectively avoided and efficiency is greatly improved.

Thirdly, as shown in Table 3, ANN has great potential and value in reducing risks in the current complex and uncertain environments in CM. The ANN model can identify and evaluate risks in the new environment by capturing the interdependence between accidents and their causes in historical data, which effectively avoids the limitations of traditional risk analysis, such as the vagueness and subjectivity of expert experience. In the case of high uncertainty, ANN has been predominantly adopted for risk analysis in terms of finance [64], safety [77], contract [46], and quality [136]. Research on project dispute claim risk prediction, optimal risk allocation in PPP projects, risk analysis for BOT project contracts, early-warning for site work risk and bid selection has been carried out to improve the level of risk management. Therefore, ANN-based risk analysis can provide auxiliary and predictive insights on key issues, helping project managers to quickly determine the priority of possible risks and to determine positive actions, such as simplifying the work site operation, adjusting personnel arrangements, and ensuring that the project is carried out on time and on budget, rather than relying on risk mitigation measures.

*4.2. Challenges and Future Directions for ANN in CM*

In the future, CM will experience rapid digital transformation and ANN will also attract more and more attention, both in academic research or in practical applications. Despite the potential opportunities and benefits accruing from ANN in CM, there are still some challenging issues that deserve further study. Only by clarifying these challenges can effective countermeasures be proposed more targeted and effectively. The following four important challenges and directions to further tackle a diversity of existing issues in laborious, complex, or even dangerous tasks in CM are put forward.

### 4.2.1. More Collaboration Is Essential for Rapid Progress in ANN in CM

Lack of collaboration is a symptom of lower research productivity [25] and strong collaborative relationships should be fostered to make better research progress [163]. According to the co-authorship analysis (Figure 2) and the collaboration network of countries/regions (Figure 3), although more and more researches have been conducted on ANN in CM in the past 20 years, as a new technology in the digital transformation of the construction industry, collaboration is still in its infancy. Except for a few scholars such as Minyuan Cheng, Hsingchih Tsai, Xuefeng Zhao and Mingyuan Zhang shown in Figure 2 and countries/regions such as the People's Republic of China and USA, which are active in this field and have moderate cooperation, the research in this field needs extensive attention, promotion and cooperation from more scholars from all over the world and different domains such as mathematics and computing as well as CM. Cooperation among them should be strengthened to enable progress in this cross-disciplinary area.

### 4.2.2. The System Design and the Platform Establishing for ANN in CM Has Not Yet Begun

According to Table 2 and Figure 4, among the existing 112 articles, some articles focus on algorithm optimization which can be verified by the keywords including genetic algorithm, regression analysis, fuzzy logic, etc., some focus on function realization with keywords such as prediction, estimation, identification, decision, cluster analysis etc., while some focus on problem solving with keywords such as cost, performance, safety, productivity, risk etc. These findings indicate that the current research is mainly focused on specific applications, and that research on system design has not attracted enough attention. Systematic design and platform building are crucial to the spread and application of new technologies [37]. For example, as to BIM, there are many researches on framework and platforms [164], and for manufacturing industry, the system optimization based on AI has gradually become a prominent research trend [165]. However, systematic research and platform building research on ANN in CM have not been started yet. In the future, data preparation, model optimization and application, system design and platform setup deserve further discussion which is critical to digital transformation.

### 4.2.3. Research Focused on Different Stakeholders as Well as the Data Sharing among Them Is Still Missing

There are many participants in the construction projects, and each have different data types and application demands [126]. Therefore, the research on ANN in CM aimed at specific stakeholders such as the owners and the contractors as well as the consultants has not yet been addressed sufficiently in practical application [107]. Meanwhile, among the construction project's key five objectives, although cost, schedule, quality, environment and safety have attracted some specific research, research on the whole life cycle of the project is still insufficient [68]. For example, systematic research is needed on what data stakeholders will have access to, and what can they do with it separately targeting the government, the owner, the contractor as well as consultants. Furthermore, the data sharing among all the stakeholders should be discussed to maximize the value of the data and minimize the harm of information asymmetry [107]. Therefore, data collection, processing, storage and application targeting specific stakeholder are worth systematic research in the future.

### 4.2.4. Data Collection Is the Key of ANN in CM

Current research has focused on the development and application of ANN models but ignored the input data preparation. According to Bilal et al. [166], the whole data mining work usually takes about 80% of the time to prepare the data. There are many data sources for ANN application in CM, and Figure 8 shows six types of data sources and frequency distribution of all articles. The most two extensive data sources are historical project files and online databases, both of which are important in mining historical engineering data and have the disadvantages of one-time data collection. Only a few scholars realize the importance of establishing a database that can be dynamically updated in real time for

a long time, such as Baalousha and Celik [99] who created a data warehouse so as to integrate all kinds of cost related information for cost estimation. Meanwhile, because of the complexity of CM, errors in data collection, processing and application are inevitable in most cases. High quality data is the prerequisite for the success of big data projects. Problems such as null value, misleading value, outlier value and non-standard value make the application of ANN extremely challenging. Although previous scholars did consider this problem and have completed data standardization [55], they did not conduct comprehensive integration and structured data quality analysis. Future research could focus on efficient and standardized data collection from massive historical engineering files through information technology, and strengthen the development and application of databases to realize sustainable data updating and data mining utilization.

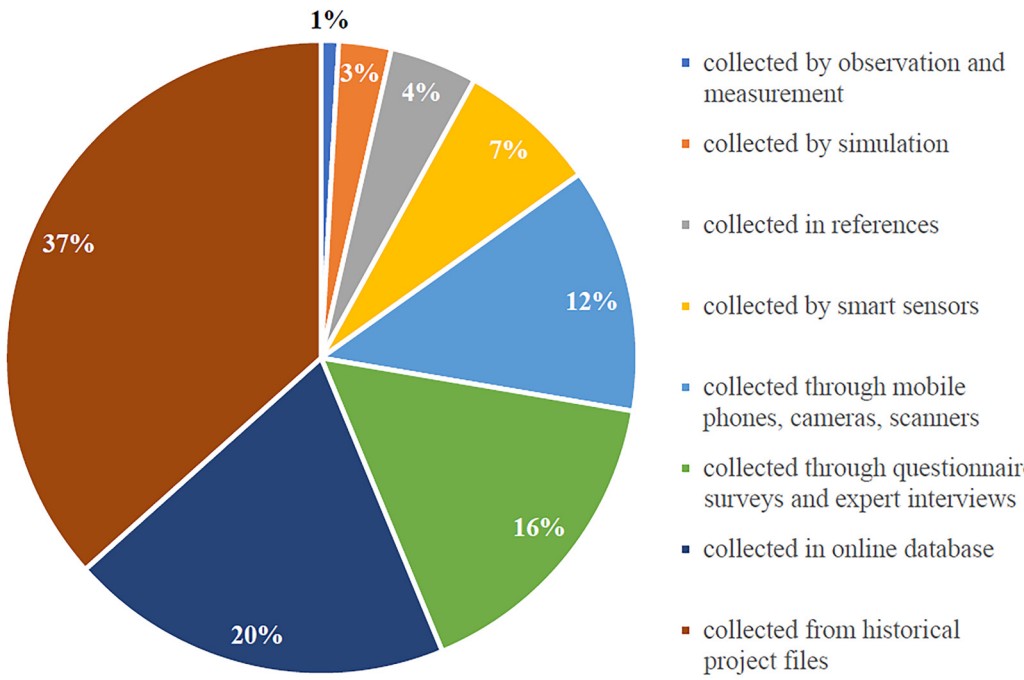

**Figure 8.** Methods of data collection of ANN application in CM.

*4.3. Limitations of This Paper*

Although some interesting findings have been made, there are still some shortcomings in this article that will be discussed in this section. Firstly, only a scientometric analysis is carried out in this paper. In order to get more detailed information on current research on ANN in CM, a content analysis can be carried out in the future. Furthermore, since the results and discussions are based on the findings of previous studies, the resulting theoretical framework should be validated and tested in future empirical studies. Moreover, although the comprehensiveness of the selected literature has been ensured to the extent possible, more searches of other database and additional keywords can be added in the future researches.

**5. Conclusions**

As an adaptive, model-free data mining method, ANN is one of the most promising data processing techniques in AEC, and has been increasingly used in CM. However, in the past 20 years, few papers have attempted to provide a comprehensive review of the existing literature on ANNs in CM. This study has analyzed a selection of 112 articles published in 7 high-quality journals between 2000 and 2020, and has conducted a comprehensive and structured review of ANN in CM. Through scientometric analysis, the review visualized the authors and countries/regions, main research interests and trends, providing a basis for further understanding of the application of ANN in CM. Challenges and future directions

were put forward to provide references for future research. At present, there is still a lack of systematic research and sufficient attention to the application of ANN in CM. ANN applications still face many challenges such as data collection, cleaning and storage, the collaboration of different stakeholders, researchers and countries/regions, as well as the systematic design for the platform. More research is still needed in these fields so as to truly achieve intelligent CM based on ANN. The uniqueness of this paper is that it limited the research subject to research on ANN in CM rather than a broader field which is very important to clarify the current research status in the field of construction management. Despite all the contributions, this review has some limitations. Future directions should focus more on content analysis, example validation, and increased retrieval of databases and keywords.

**Author Contributions:** Methodology, H.X. and S.L.; software, H.X. and H.L.; validation, R.C. and M.P.; formal analysis, H.X.; writing—original draft preparation, H.X; writing—review and editing, R.C.; visualization, H.L. and S.L.; supervision, M.P. and R.J.W.; project administration, N.D. and J.Z. All authors have read and agreed to the published version of the manuscript.

**Funding:** This research received no external funding.

**Institutional Review Board Statement:** Not applicable.

**Informed Consent Statement:** Not applicable.

**Data Availability Statement:** Some or all data, models, or code that support the findings of this study are available from the corresponding author upon reasonable request.

**Conflicts of Interest:** The authors declare no conflict of interest.

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
