# Peer review of "Application of Artificial Neural Networks in Construction Management: A Scientometric Review"

_buildings, doi:10.3390/buildings12070952_

Round 1

Reviewer 1 Report

It appears as though some disjointed sentences were combined and presented in the introduction. The introduction’s logic is a little difficult to follow, and several statements are irrelevant. I strongly advise the authors to rewrite the entire introduction and present it in a more systematic manner. I propose that they begin this section by addressing a major issues such as urbanisation and the growing demand for construction. They demonstrate the critical role that construction management plays in the construction industry. Finally, they can discuss the various methods and approaches that can be used to facilitate decision-making,.... This means they can assert that traditional methods have some drawbacks and that, as a result, artificial intelligence is currently being used. Which of these methods is an ANN capable of X, Y, Z...? Given the importance of this subject, this study fills in the gaps and paves the way for future research...... Numerous these issues were addressed in the introduction but in a very haphazard manner.

This appears to be a reviewer’s comment in the first paragraph of Section 2. Am I correct? This isn’t part of your paper.

According to the authors, they only used WOS. While WOS covers a limited number of papers, this is one of the paper’s drawbacks. WOS and Scopus are almost always used in conjunction.

Between Lines 125 and 129, the following statements are completely unacceptable. This results in the neglect of a number of excellent papers. Numerous examples exist that do not meet these criteria.

“The following retrieval code was 132 adopted, and the search was conducted using the ‘topic’ in the Web of Science. (neural network) AND (construction management), (neural network) AND (engineering 134 management) “ this section of the paper exemplifies the major flaw in the research methodology that renders the study unreliable. This strategy undoubtedly overlooks a large number of papers. The authors define construction management in the following manner. Certain papers may include the terms earned value and ANN in their titles, which this strategy cannot capture. Additionally, there are numerous papers in the literature that use ANNs to forecast costs, budgets, and the reliability of equipment used in construction projects......but do not specifically mention construction management in their titles. As a result, this research methodology is completely doubted.

Due to the fact that this paper is woefully inaccurate, and the research methodology cannot guarantee reliable results, I am unable to recommend it even for a major revision at all, and the authors should conduct another research. I must reject it.  

Author Response

Detailed Response to Reviewers 1

Reviewer#1:

Due to the fact that this paper is woefully inaccurate, and the research methodology cannot guarantee reliable results, I am unable to recommend it even for a major revision at all, and the authors should conduct another research. I must reject it.

Response to Reviewer#1

Thank you for your extremely harsh criticism and correction. We have substantially revised the article according to your opinions, and made explanations and responses to some core issues, hoping to make you change your mind and reconsider this article. The following is a list of responses to each of your comments and suggestions: (The italics are the original text in the manuscript):

Q1: It appears as though some disjointed sentences were combined and presented in the introduction. The introduction’s logic is a little difficult to follow, and several statements are irrelevant. I strongly advise the authors to rewrite the entire introduction and present it in a more systematic manner. I propose that they begin this section by addressing a major issues such as urbanisation and the growing demand for construction. They demonstrate the critical role that construction management plays in the construction industry. Finally, they can discuss the various methods and approaches that can be used to facilitate decision-making,.... This means they can assert that traditional methods have some drawbacks and that, as a result, artificial intelligence is currently being used. Which of these methods is an ANN capable of X, Y, Z...? Given the importance of this subject, this study fills in the gaps and paves the way for future research...... Numerous these issues were addressed in the introduction but in a very haphazard manner.

A1: Thank you for your valuable advice. We have substantially revised the introduction to ensure logical coherence and ease of reading. The writing ideas and structure of the revised Introduction are as follows: (1) Construction management, which is very important for the construction industry, is in urgent need of updating its methods and concepts. (2) Data has become the focus of construction management in the future. However, data utilization in the construction industry is still insufficient, because it is difficult for traditional technologies to effectively utilize large volume data in the construction industry. (3) As a new intelligent technology, ANN is one of the most promising technologies of data mining in construction management, which has the many advantages. (4) What is the current status of literature review about ANN application in CM, what are the shortcomings, and what is the importance of this study. (5) Introduce the purpose of this paper. A part of the rewritten Introduction is attached as follow and we hope the rewritten introduction will be appreciated by you and other readers.

Introduction (Line 41, Page 1 to Line 56, Page 2): The characteristics of high investment, long period and high uncertainty make construction management become an indispensable part of modern construction industry [1, 2]. The urgent need of upgrading and transformation of construction industry also drives the renewal of construction management concepts and methods [3-5]. In the data-intensive industry, data, which can significantly improve the performance of CM, is becoming the key resource in the future construction industry [6, 7]. Nevertheless, data application in CM has been considered relatively conservative [1]. It is difficult to analyze and process the large volume data of the construction industry with tradition-al methods, so that a large amount of data is shelved and wasted [8]. The digitization report released by McKinsey indicated that the construction industry is currently one of the worst performing in digitalization, which maybe is the main reason for decades of persistently low productivity in the construction industry [9, 10]. Therefore, in the digitalization era, it is of great significance for the construction industry to use intelligent technology to process large volume data in CM and obtain knowledge hidden in the data for assist decision making [11]. Furthermore, one of the most promising technologies is artificial neural network (ANN) [12].

Q2: This appears to be a reviewer’s comment in the first paragraph of Section 2. Am I correct? This isn’t part of your paper.

A2: We are very sorry that we did not do a good job of manuscript review at the beginning, so that we left the requirements of submission documents in the body. At present, we have carefully checked the whole manuscript to ensure that such mistakes will not happen again. Meanwhile, a careful proofreading of the article has been carried out to avoid English errors and typos.

Q3: According to the authors, they only used WOS. While WOS covers a limited number of papers, this is one of the paper’s drawbacks. WOS and Scopus are almost always used in conjunction.

A3: Thank you very much for your questioning of this article, which also contributed to the improvement of this article. Indeed, many researchers use both WOS and Scopus as databases of literature sources. However, it is also considered reasonable to use one of these databases as a separate source. The following is a partial list of literature review that uses only one database as a source: [1-13]. Among them, the review that only uses WOS as the data source is [1, 2, 6-10, 12, 13]. The review that uses WOS as database to study the application of new technology in CM is [1, 2, 7, 10]. Therefore, it is reasonable to select only WOS as the literature source database in this paper. There may be a few articles that are not selected, but it will not have a significant impact on the final result. Furthermore, using only a single database have been added into the research limitations of this paper.

Q4: Between Lines 125 and 129, the following statements are completely unacceptable. This results in the neglect of a number of excellent papers. Numerous examples exist that do not meet these criteria. “The following retrieval code was adopted, and the search was conducted using the ‘topic’ in the Web of Science. (neural network) AND (construction management), (neural network) AND (engineering management) “ this section of the paper exemplifies the major flaw in the research methodology that renders the study unreliable. This strategy undoubtedly overlooks a large number of papers. The authors define construction management in the following manner. Certain papers may include the terms earned value and ANN in their titles, which this strategy cannot capture. Additionally, there are numerous papers in the literature that use ANNs to forecast costs, budgets, and the reliability of equipment used in construction projects......but do not specifically mention construction management in their titles. As a result, this research methodology is completely doubted.

A4: Thank you also for your serious questions about this article, which contributed to the progress of this article. Firstly, according to the key search words definition of some similar reviews, the key search words retrieval method (“Research field” AND “Research technology”) adopted in this paper is reasonable. For details, please refer to [1, 2, 4, 6, 7, 11, 14, 15]. For example, as in Liu, Le, Hu, Xia, Skitmore and Gao [4], key search words are defined as “system dynamics” and “construction”. Chen and Pan [1] also adopts a similar retrieval method and sets the keywords of topic as “fuzzy” AND “WSM”. Similarly, Araújo, Pereira Carneiro and Palha [2] has simply set their words used in the searches as the “Construction management” and “Sustainab*”. Moreover, this paper adopts the “topic” search rather than the “title” search, which the former can cover most of the content of the article so as not to be misjudged. There may be a small number of studies that are not considered. But in a large volume of literature analysis, these few documents will not have an unacceptable impact on the final analysis results. In the final limitations of this paper, the uncertainty caused by key search words is also raised. We sincerely hope that you will reconsider our article.

Reference

  1. L. Chen, et al., Review fuzzy multi-criteria decision-making in construction management using a network approach, Applied Soft Computing 102 (2021).
  2. A.G. Araújo, et al., Sustainable construction management: A systematic review of the literature with meta-analysis, Journal of Cleaner Production 256 (2020).
  3. W.M. Jayantha, et al., Bibliometric analysis of hedonic price model using CiteSpace, International Journal of Housing Markets and Analysis 13(2) (2019) 357-371.
  4. M. Liu, et al., System Dynamics Modeling for Construction Management Research: Critical Review and Future Trends, Journal of Civil Engineering and Management 25(8) (2019) 730-741.
  5. A. Asadzadeh, et al., Sensor-based safety management, Automation in Construction 113 (2020).
  6. E.M.A.C. Ekanayake, et al., Mapping the knowledge domains of value management: a bibliometric approach, Engineering, Construction and Architectural Management 26(3) (2019) 499-514.
  7. H. Wang, et al., Integration of BIM and GIS in sustainable built environment: A review and bibliometric analysis, Automation in Construction 103 (2019) 41-52.
  8. W. Hu, et al., Research progress on ecological models in the field of water eutrophication: CiteSpace analysis based on data from the ISI web of science database, Ecological Modelling 410 (2019).
  9. X. Liu, et al., Visualized analysis of knowledge development in green building based on bibliographic data mining, The Journal of Supercomputing 76(5) (2018) 3266-3282.
  10. L. Lima, et al., Sustainability in the construction industry: A systematic review of the literature, Journal of Cleaner Production 289 (2021).
  11. P. Martinez, et al., A scientometric analysis and critical review of computer vision applications for construction, Automation in Construction 107 (2019).
  12. F. Xiao, et al., Knowledge Domain and Emerging Trends in Organic Photovoltaic Technology: A Scientometric Review Based on CiteSpace Analysis, Front Chem 5 (2017) 67.
  13. G.-L. Jia, et al., Review of Urban Transportation Network Design Problems Based on CiteSpace, Mathematical Problems in Engineering 2019 (2019) 1-22.
  14. M. Ershadi, et al., Core capabilities for achieving sustainable construction project management, Sustainable Production and Consumption 28 (2021) 1396-1410.
  15. X. Hu, et al., The application of case-based reasoning in construction management research: An overview, Automation in Construction 72 (2016) 65-74.

Reviewer 2 Report

The manuscript analyzed the state-of-art journal papers published in 2000-2020 regarding application of Artificial Neural Network in the field of Construction Management using a scientometric reviewing methodology. The detailed analysis results based on the scientometric review are comprehensive and provide the current states of application of ANN in the field of CM. Further, the suggested future directions for further studies would provide valuable guidances for those who are interested in application of ANN to CM. The presented English is neat and easy to understand and the context is well-organized although there are some minor grammar errors. 

The following comments are offered for the authors’ consideration. 

  1. (General) : Although the methodology for scientometric analysis is simply provided in the manuscript, detailed processes of treating the collected data or using VOSviewer and Citespace are missing. It would be beneficial to readers who are interested in the field of scientometric analysis.
  2. (Page 1 Line 34-35): Abstract should summarize the whole contents of the manuscript. However, “the valuable findings” seem missing in the manuscript. Please briefly describe what are your findings in Abstract.
  3. (Page 3 Line 105-110): The whole phrase is confusing, and the reviewer cannot find any coherence to your next phrase with it. It seems a submission guideline was mistakenly inserted. If so, please delete the corresponding phrase.
  4. (Page 5 Line 179): Please explain what the network density means for better understanding of readers. 
  5. (Page 5 Line 186-187): The authors revealed that the most important scholars in the application of ANN in CM based on the number of articles, assuming that “the number of articles indicates the importance of scholars”. Please provide such assumption to prevent unnecessary misunderstanding of readers.
  6. (Page 5 Line 206-210, Page 8 Line 277): Please provide explanation for the centrality, Modularity(Q), and Mean Silhouette value (MS) for better understanding of readers.
  7. (Page 14 Line 461-475): Unlike 4.2.1, 4.2.2, 4.2.4, evidence for the authors’ opinion seems very weak in 4.2.3 without any data nor citation. Please provide any evidence for “lack of data sharing”.  

Author Response

Detailed Response to Reviewers 2

Reviewer#2:

The manuscript analyzed the state-of-art journal papers published in 2000-2020 regarding application of Artificial Neural Network in the field of Construction Management using a scientometric reviewing methodology. The detailed analysis results based on the scientometric review are comprehensive and provide the current states of application of ANN in the field of CM. Further, the suggested future directions for further studies would provide valuable guidances for those who are interested in application of ANN to CM. The presented English is neat and easy to understand and the context is well-organized although there are some minor grammar errors.

Response to Reviewer#2

Thanks for your comments on our paper. Your comments are all valuable and very helpful for revising and improving our paper, as well as the important guiding significance to our paper. We have revised our paper according to your comments as following (The italics are the original text in the manuscript):

Q1: (General): Although the methodology for scientometric analysis is simply provided in the manuscript, detailed processes of treating the collected data or using VOSviewer and Citespace are missing. It would be beneficial to readers who are interested in the field of scientometric analysis.

A1: Thank you for your valuable advice. We have made some adjustments to the structure of the paper and added detailed processes of treating the collected data or using VOSviewer and Citespace (see in Section 2.2 Introduction and process of scientometric analysis, Line 155 to 172, Page 4). The revised manuscript as follows:

In Section 2.2 (Line 155 to 172, Page 4): Whether VOSviewer or Citespace, its main process can be summarized as follows :(1) importing literature data from WOS into software for visualization;(2) Figures presentation and optimization of different aspects according to software functions;(3) In-depth Scientometric analysis of the adjusted figures. After screening mentioned above, each piece of data is downloaded from WOS as a full-record text format. In total, 112 valid publications were extracted. Then, the output file format of the re-name is ‘download_*.txt’ and is imported into VOSviewer and Citespace for format conversion. Visualizations are generated and can be converted to different views through different functions of the software. After adjusting different visualizations of input data to make them easier to read and analyze, the scientometric analysis is be-ginning, which includes the following 4 aspects in this paper. First, through co-author network analysis, core research groups and their cooperative relationships were identified. Second, with network analysis of participating countries/regions the most influential countries/regions who are particular active on ANN in CM and the collaborations among them were described. Third, towards the key-words, network analysis is conducted to discern the main research interests and the hot topics on ANN in CM. Fourth, with the timeline visualization and citation bursts, keywords evolution network affords the trends and changes. Finally, network analysis of the co-citation reference was carried out to mine the most representative literature in this filed.

Q2: (Page 1 Line 34-35): Abstract should summarize the whole contents of the manuscript. However, “the valuable findings” seem missing in the manuscript. Please briefly describe what are your findings in Abstract.

A2: It is true that the original abstract is not accurate and rigorous enough and therefore has been rewritten. The contents and contributions have been supplemented with generalized languages as follows because of the 200-word limitation.

Abstract (Line 22 to 36, Page 1): As a powerful artificial intelligence tool, the Artificial Neural Network (ANN) has been increasingly applied in the field of construction management (CM) during the last few decades. However, few papers attempted to draw up a systematic commentary to appraise the state-of-the-art research on ANN in CM except the one published in 2000. In this paper, a scientometric analysis was adopted to comprehensively analyze 112 related articles retrieved from seven selected authoritative journals published between 2000 and 2020. Through co-authorship network, collaboration network of countries/regions, co-occurrence network of keywords, timeline visualization of keywords together with the strongest citation burst, the active research authors, countries/regions, main research interests together with their evolution trend and collaboration ships in the past 20 years etc. are comprehensively summarized and analyzed. This paper found that there is still a lack of systematic research and sufficient attention on the application of ANN in CM. Furthermore, ANN applications still face many challenges such as data collection, cleaning and storage, the collaboration of different stakeholders, researchers and countries/regions, as well as the systematic design for the platform. The findings are valuable to both the researcher and industry practitioners who are committed to ANN in CM.

Q3: (Page 3 Line 105-110): The whole phrase is confusing, and the reviewer cannot find any coherence to your next phrase with it. It seems a submission guideline was mistakenly inserted. If so, please delete the corresponding phrase.

A3: We are very sorry that we did not do a good job of manuscript review at the beginning, so that we left the requirements of submission documents in the body. At present, we have carefully checked the whole manuscript to ensure that such mistakes will not happen again. Meanwhile, a careful proofreading of the article has been carried out to avoid English errors and typos.

Q4: (Page 5 Line 179): Please explain what the network density means for better understanding of readers.

A4: According to your valuable advice, the explanation of network density has been added to the revised manuscript, which can be seen as following:

In Section 3 (Line 183 to 186, Page 5): As can be seen from Figure 2, there are 253 nodes, 287 links, and the network density is 0.0101. The typical value for network density is between 0 and 1. Especially low net-work density, even close to 0, indicates that the authors in the network are not closely connected [147].

Q5: (Page 5 Line 186-187): The authors revealed that the most important scholars in the application of ANN in CM based on the number of articles, assuming that “the number of articles indicates the importance of scholars”. Please provide such assumption to prevent unnecessary misunderstanding of readers.

A5: Thank you for your valuable advice. Initially, this assumption came from description of Hu, et al. [1] to the core author. The article describes it this way: In terms of co-occurrence frequency, the top two authors are Arhonditsis, G.B. and Mooij, W.M. with co-occurrence times of 21 and 17, respectively. These represent the most important scholars in the application of ecological models for eutrophication and lay the academic foundation in this field. However, after your correction, we realize that the same assumption may not be made in the field of ANN and CM. Therefore, this article has changed the description as follow to avoid unnecessary misunderstanding:

In Section 3.1(Line 190 to 192, Page 5): Among them, the cluster with MINYUAN CHENG is the core research team. They represent the important scholars in the application of ANN in CM and can offer highly individualized scientific research information to other researchers in this field.

Q6: (Page 5 Line 206-210, Page 8 Line 277): Please provide explanation for the centrality, Modularity(Q), and Mean Silhouette value (MS) for better understanding of readers.

A6: Thank you for pointing out this serious problem. The meanings of these indicators are not clearly described in the original text. Therefore, we have added a clear description of these indicators and supporting literature in the revised manuscript for readers to understand.

The description of centrality in Section 3.2 (Line 210 to 213, Page 6): The betweenness centrality is an important index in Citespace. Freeman [148] noted that the betweenness centrality could be calculated by the ratio of the shortest path between two nodes to the sum of all such shortest paths. The greater the betweenness centrality, the higher its importance.

The description of Modularity(Q), and Mean Silhouette value (MS) in Section 3.3.1 (Line 210 to 213, Page 6): Usually, the Modularity (Q value) and Mean Silhouette (MS) value is used to evaluate the clustering effect. Generally speaking, Q value is generally within the interval of [0,1). Q > 0.3 means that the community structure divided is significant. A cluster's silhouette value ranges from −1 to 1 and assesses the uncertainty involved in defining the cluster's nature. A value of 1 signifies that the cluster is perfectly isolated [21]. As shown in Figure 5, the network has Modularity (Q = 0.581>0.3),which represents that the strength of dividing the network into clusters is high with dense links amid nodes within clusters. And the MS value is 0.363, which indicated the homogeneity of the clusters is not high. This MS value shows that although these studies in the network in each cluster may be consistent in exploring similar issues, these address different issues in fact [21].

Q7: (Page 14 Line 461-475): Unlike 4.2.1, 4.2.2, 4.2.4, evidence for the authors’ opinion seems very weak in 4.2.3 without any data nor citation. Please provide any evidence for “lack of data sharing”.

A7: Thank you for spotting this terrible problem. The viewpoints of this paper are both supported by data and literature. However, when the submission format was changed, the supporting literature in 4.2.3 was deleted by mistake without being discovered. At present, the references in this paper have been comprehensively checked to ensure that similar problems will not occur. This section is updated as follows:

In section 4.2.3 (Line 481 to 493, Page 14): There are many participants in the construction projects, and each have different data types and application demands [127]. Therefore, the research of ANN in CM aimed at specific stakeholders such as the owners, the contractors as well as the con-sultants has not yet been addressed which is of significance in practical application [108]. Meanwhile, as the project's most concerned five objective, although cost, sched-ule, quality, environment and safety have attracted some specific research, research on the whole life cycle of the project is still insufficient [69]. For example, what data will they have access to, and what can they do with it separately targeting the government, the owner, the contractor as well as the consultant should be further researched systematically. Furthermore, the data sharing among all the stakeholder should also be discussed to maximize the value of the data and minimize the harm of information asymmetry [108]. Therefore, the data collection, processing, storage and application targeting specific stakeholder is worth systematic research in the future.

Reference

  1. W. Hu, et al., Research progress on ecological models in the field of water eutrophication: CiteSpace analysis based on data from the ISI web of science database, Ecological Modelling 410 (2019).

Reviewer 3 Report

Dear the authors,

Thank you for the opportunity to review your paper.

This paper is very interesting, dealing with valuable academic/practical issues.

However, after some editorial changes and further explanation, the paper is worth to be published.

1. The abstract should also include essential aspects of the study methods. The abstract must be included clear description of study methodology and conclusion obtained besides the aim of the study. The abstract should be re-written so that it encompasses summaries of the most important parts of the study results and authors' arguments.

2. There is no limitation of this study. Therefore, the discussion chapter should be added and it should be described the construction industry implications and limitations.

3. The 112 papers should be added to the references list.

4. All tables and figures in this paper should be explained in detail in an easy-to-understand manner for the reader.

Author Response

Detailed Response to Reviewers 3

Reviewer#3:

Comments to the Author

Thank you for the opportunity to review your paper. This paper is very interesting, dealing with valuable academic/practical issues. However, after some editorial changes and further explanation, the paper is worth to be published.

Response to Reviewer#3

Thanks for your comments on our paper. Your comments are all valuable and very helpful for revising and improving our paper, as well as the important guiding significance to our paper. We have revised our paper according to your comments (The italics are the original text in the manuscript):

Q1: The abstract should also include essential aspects of the study methods. The abstract must be included clear description of study methodology and conclusion obtained besides the aim of the study. The abstract should be re-written so that it encompasses summaries of the most important parts of the study results and authors' arguments.

A1: It is true that the original abstract is not accurate and rigorous enough and therefore has been rewritten. The contents and contributions have been supplemented with generalized languages as follows because of the 200-word limitation.

Abstract (Line 22 to 36, Page 1): As a powerful artificial intelligence tool, the Artificial Neural Network (ANN) has been increasingly applied in the field of construction management (CM) during the last few decades. However, few papers attempted to draw up a systematic commentary to appraise the state-of-the-art research on ANN in CM except the one published in 2000. In this paper, a scientometric analysis was adopted to comprehensively analyze 112 related articles retrieved from seven selected authoritative journals published between 2000 and 2020. Through co-authorship network, collaboration network of countries/regions, co-occurrence network of keywords, timeline visualization of keywords together with the strongest citation burst, the active research authors, countries/regions, main research interests together with their evolution trend and collaboration ships in the past 20 years etc. are comprehensively summarized and analyzed. This paper found that there is still a lack of systematic research and sufficient attention on the application of ANN in CM. Furthermore, ANN applications still face many challenges such as data collection, cleaning and storage, the collaboration of different stakeholders, researchers and countries/regions, as well as the systematic design for the platform. The findings are valuable to both the researcher and industry practitioners who are committed to ANN in CM.

Q2: There is no limitation of this study. Therefore, the discussion chapter should be added and it should be described the construction industry implications and limitations

A2: Thank you for your valuable advice. The limitations of this paper have been added to the Section 4. Discussion. The revised manuscript is shown below:

In Section 4.2.3 (Line 516 to 524, Page 15): Although some interesting findings have been made, there are still some short-comings in this article that will be discussed in this section. Firstly, only a scientometric analysis is carried out in this paper. In order to get more detailed information on cur-rent research on ANN in CM, a content analysis can be carried out in the future. Furthermore, since the results and discussions are based on the findings of previous studies, the resulting theoretical framework should be validated and tested in future empirical studies. Moreover, although the comprehensiveness of the selected literature has been ensured to the extent possible, more searches of other database and additional keywords can be added in the future researches.

Q3: The 112 papers should be added to the references list.

A3: According to your valuable comments, the 112 selected paper has been added in the Table 1, which be considered as the appropriate insertion position. The revised Table 1 can be seen on Line 136, Page 4.

Q4: All tables and figures in this paper should be explained in detail in an easy-to-understand manner for the reader.

A4: Thank you for your valuable advice. We have carefully checked the description of each table and picture according to your opinion, and added indicator explanation to ensure the readability of the article. The following are some examples of added and revised descriptions and indicator interpretations:

In Section 3.1 (Line 178 to 186, Page 5): In this paper, the co-authorship network was generated in Citespace and the collabo-ration map is presented in Figure 2. The size of the node indicates the number of the articles published by the author, and the connecting line indicates the collaboration relationship among them, and the color of the line indicates the author in the same cluster. Publication dates from past to present are shown in a transition from cool to warm color. As can be seen from Figure 2, there are 253 nodes, 287 links, and the network density is 0.0101. The typical value for network density is between 0 and 1. Especially low net-work density, even close to 0, indicates that the authors in the network are not closely connected [147].

In Section 3.2 (Line 201 to 204, Page 5): Figure 3 shows the network of citing countries/regions, this network contains 29 nodes and 33 links. The size of a node represents the total number of published articles in the 112 articles, and the thickness of the links indicates the levels of the cooperative relationships.

In Section 3.3.1 (Line 246 to 248, Page 7): Table 2 summarizes the keyword occurrences and node strength of each individual. The links are the number of linkages between a given node and others, while the total link strength reflects the total strength linked to a specific item.

In Section 3.3.2 (Line 285 to 296, Page 8): CiteSpace automatically selects a label for each cluster based on titles, keyword, and abstracts of the articles in each cluster. Usually, the Modularity (Q value) and Mean Silhouette (MS) value is used to evaluate the clustering effect. Generally speaking, Q value is generally within the interval of [0,1). Q > 0.3 means that the community structure divided is significant. A cluster's silhouette value ranges from −1 to 1 and assesses the uncertainty involved in defining the cluster's nature. A value of 1 signifies that the cluster is perfectly isolated [21]. As shown in Figure 5, the network has Modu-larity (Q = 0.581>0.3),which represents that the strength of dividing the network into clusters is high with dense links amid nodes within clusters. And the MS value is 0.363, which indicated the homogeneity of the clusters, is not high. This MS value shows that although these studies in the network in each cluster may be consistent in exploring similar issues, these address different issues in fact [21].

In Section 3.3.2 (Line 331 to 334, Page 10): Figure 6 shows the top 25 keywords with strongest citations bursts from 2000 to 2020. The light green line indicates the range of literature years reviewed, while the orange line indicates the duration of a citation burst event.

In Section 3.4 (Line 355 to 360, Page 11): CiteSpace was used to analyze the documents cited in 112 records, Figure 7 shows the detailed outcome of document co-citation analysis, i.e., co-citation network including 497 nodes and 1485 links. Each link means a co-citation relationship between the two corresponding articles while the size of the font represents the co-citation frequency of these documents. The node documents were among cited documents and were not necessarily included in the 112 retrieved articles.

Round 2

Reviewer 1 Report

The answer of the authors to questions 4 and 5 are not acceptable at all. the research methodology of this research has many issues.